# African Swine Fever Virus Manipulates the Cell Cycle of G0-Infected Cells to Access Cellular Nucleotides

**DOI:** 10.3390/v14081593

**Published:** 2022-07-22

**Authors:** Hranush R. Avagyan, Sona A. Hakobyan, Arpine A. Poghosyan, Nane V. Bayramyan, Hranush H. Arzumanyan, Liana O. Abroyan, Aida S. Avetisyan, Lina A. Hakobyan, Elena M. Karalova, Zaven A. Karalyan

**Affiliations:** 1Laboratory of Cell Biology and Virology, Institute of Molecular Biology of NAS RA, Yerevan 0014, Armenia; a.avagian@yahoo.com (H.R.A.); 777sona7@gmail.com (S.A.H.); arpi.poghosyan21@gmail.com (A.A.P.); naneramaz@mail.ru (N.V.B.); hharzumanyan@gmail.com (H.H.A.); liana.abroyan@gmail.com (L.O.A.); a.avetis@mail.ru (A.S.A.); lina.hakobyan@gmail.com (L.A.H.); hatussili@yahoo.com (E.M.K.); 2Experimental Laboratory, Yerevan State Medical University after Mkhitar Heratsi, Yerevan 0025, Armenia

**Keywords:** ASFV, G0 cells, G1 phase of cell cycle, S phase of cell cycle, cell cycle activation, cellular DNA synthesis

## Abstract

African swine fever virus manipulates the cell cycle of infected G0 cells by inducing its progression via unblocking cells from the G0 to S phase and then arresting them in the G2 phase. DNA synthesis in infected alveolar macrophages starts at 10–12 h post infection. DNA synthesis in the nuclei of G0 cells is preceded by the activation of the viral genes K196R, A240L, E165R, F334L, F778R, and R298L involved in the synthesis of nucleotides and the regulation of the cell cycle. The activation of these genes in actively replicating cells begins later and is less pronounced. The subsequent cell cycle arrest at the G2 phase is also due to the cessation of the synthesis of cellular factors that control the progression of the cell cycle–cyclins. This data describes the manipulation of the cell cycle by the virus to gain access to the nucleotides synthesized by the cell. The genes affecting the cell cycle simply remain disabled until the beginning of cellular DNA synthesis (8–9 hpi). The genes responsible for the synthesis of nucleotides are turned on later in the presence of nucleotides and their transcriptional activity is lower than that during virus replication in an environment without nucleotides.

## 1. Introduction

African swine fever virus (ASFV) is the only species in the genus Asfivirus, family Asfarviridae, and order Asfuvirales. It is a large double-stranded DNA virus [1,2].

Many DNA viruses interfere with the cell cycle regulatory machinery. Some viruses, following infection, require de novo synthesis of deoxynucleotides to stimulate G1 to S phase cell cycle transition in cells [3]. Other viruses (like Herpesviruses) can cause cell cycle arrest to limit competition between the virus and host for cellular DNA replication resources. However, in most cases, the manipulation of viruses by the host cell cycle contributes to a favorable cellular environment for viral replication [4]. 

ASFV encodes up to 200 polypeptides that can have complex and subtle interactions with the host cell to avoid the host defenses. These viral proteins promote the replication of ASF virus in infected cells and the subsequent spread of the virus for further infections. However, there is still a lack of information on the role of many ASFV-encoded proteins in infected host cells. As cell cycle regulation is usually altered in continuous cell lines, the effects of viral replication on it may not necessarily be the same during natural infection in G0 cells (porcine macrophages) and in actively proliferating cells [5]. Thus, enzymes which are involved in nucleotide metabolism are non-essential for virus replication in dividing tissue culture cells, but their deletion reduces virus replication in macrophages [6,7]. Thus, we can assume different transcriptional activity of ASFV in various types of cells, depending on their cell cycle status and phase. According to this, we suggest investigating the transcriptional activity of the genes involved both in nucleotide and DNA synthesis in the early stage of ASFV infection. The ASFV genome has several genes involved in nucleotide and DNA synthesis. They are the thymidine kinase gene (K196R), Serine/threonine-protein kinase gene (R298L), thymidylate kinase gene (A240L), dUTP nucleotidohydrolase (E165R), and two subunits of ribonucleotide reductase (F334L, F778R) [8,9,10,11].

Previously, it was shown that the replication of ASFV can promote DNA synthesis in infected cells [12]. Moreover, G0 cells accelerate entry into both the S-phase and G2 but not into mitosis under viral infection [13]. However, there are still many questions about the reasons and mechanisms of ASFV-induced cell cycle progression and arrest. It mainly refers to ASFV, as its primary target is differentiated macrophages, which are G0 cells. In this study we investigated how ASFV manipulates the cell cycle of infected G0 cells.

## 2. Material and Methods

### 2.1. Animals

Twelve healthy pigs (Landrace breed) of the same age (three months old) and weight (30–32 kg) were used for this study. Ten pigs were infected by intramuscular injection, and two pigs were used as uninfected controls with intramuscular injection of physiological solution. 

### 2.2. Virus

The ASFV Armenia 2007 (Arm07) strain was used in all studies. The titer of ASFV for each intramuscular injection was 10^4^ 50% hemadsorbing doses (HADU50)/mL [6]. Virus titration was performed and expressed as log10 HADU50/mL for non-adapted cells. Animal experiments were carried out in accordance with the Institutional Review Board/Independent Ethics Committee of the Institute of Molecular Biology of NAS RA (reference number IRB00004079; 28 May 2018). 

The animals were divided into five groups. Two animals in each group were euthanized on the second, third, fourth, sixth and seventh day post-infection (dpi). Infections were carried out using ASFV (genotype II) distributed in the Republic of Armenia and the Republic of Georgia. The titer of ASFV for each intramuscular injection was 10^4^ 50% hemadsorbing doses (HADU50)/mL. Virus titration was done as described previously and expressed as log10 HADU50/mL for non-adapted cells [14]. 

During necropsy, the inner organs were carefully removed and fixed in a 10% buffered formalin solution (pH 7.2) for histopathology studies.

### 2.3. Alveolar Macrophage Culture

Three-month-old pigs were euthanized and their lungs were removed. Cells obtained during bronchoalveolar lavage (BAL) were suspended in sterile Hank’s balanced salt solution. They were centrifuged at 600 g for 10 min and then resuspended in RPMI 1640 with 5% fetal bovine serum (FBS) at an initial cell concentration of 3 × 10^5^ cells per mL. After incubation for 3 h at 37 °C in a humidified CO_2_ incubator, the adhered cells (porcine alveolar macrophages-PAMs) were washed three times with RPMI to remove contaminating non-adherent cells and then incubated in RPMI 1640 with 10% FBS [15].

### 2.4. PAM Infection

For in vitro experiments, ASFV was grown in primary culture of porcine alveolar macrophages (PAMs). PAMs seeded as described above were inoculated with ASFV Arm07 at 10^4^ HADU_50_/_mL_. After 1 h of adsorption at 37 °C, the infected cell monolayers were washed twice to remove unbound viruses. Afterward, a complete medium was added and analyzed at the indicated time required. All data were obtained after 1, 2, 3, 4, 5, 6, 7, 8, 9, 10, 12, 14, 16, and 24 h post infection (hpi). 

### 2.5. Nucleotides in Cultural Medium

To evaluate the effect of the presence of nucleotides on the transcriptional activity of ASFV genes, PAMs were cultivated in the presence of each base, nucleosides, and nucleotides at 1 mM concentration [16]. Nucleotides were added to PAMs (24 h) immediately before virus infection.

### 2.6. Dexamethasone in Cultural Medium

PAMs were treated with dexamethasone (1 mM, Sigma-Aldrich, Burlington, MA, USA) for 1 h and then infected by ASFV as described previously [17]. All measurements were done at 8, 9, 10, 12, 14, 16, 24, and 48 hpi.

### 2.7. Histopathological Studies

Tissue samples were fixed in 10% buffered formalin solution (pH 7.2) for a minimum of 24 h. Then, the hearts were sliced into approximately 0.8–1.0 cm thick sections and fixed in fresh formalin for at least seven more days. After the fixation, samples were dehydrated through a graded series of alcohols, washed with xylol, and embedded in paraffin wax by a routine technique for light microscopy (Microm HM 355, Thermo Fisher Scientific, Waltham, MA USA). 

Paraffin-embedded samples were cut (5 µm) and stained with a trichromic stain by [18] with the previously described modification by [19].

### 2.8. Safranin and Indigo-Picro-Carmine Staining Technique

For visualization of cell cycle deviations, specimens were treated with the combined trichromic stain (using safranin and indigo-picro-carmine). This method allows simple estimations of cell cycle kinetic parameters in cultured and biopsy specimens. The technique was used according to the author’s data; however, the fixation was carried out with 10% formalin (4-h fixation) according to [19]. This modification resulted in more pronounced differential staining of the cell cycle stages. This stain was used both for histological sections and for PAM cells.

### 2.9. Image Cytometry of the PAM/Cell Cycle Phase

All cells either uninfected or infected were fixed and stained with Feulgen-Naphthol Yellow protocol. In brief, DNA hydrolysis was performed in 5N HCl for 60 min at 22 °C. After rinsing with sulfite solution and distilled water, samples were put directly into a solution of 0.1% Naphthol Yellow S in 1% acetic acid (pH 2.8) for 30 min; they were then de-stained with 1% acetic acid three times for 0.5 min, then samples were dehydrated three times with tert-butanol and treated with xylol for 5 min [20]. 

Phases of the cell cycle were determined by examining Feulgen-stained PAM cultures with an image microspectrophotometry. The content of DNA in each sample was measured by a computer-equipped microscope-photometer SMP 05 (OPTON), and images were collected at the 575 nm wavelength. The quantity of DNA was first measured by image cytophotometry in conventional units (C.U.) [21,22]. Cytometric quantification of the DNA-staining of nuclear human cell integrated optical density (IOD) is equivalent to human DNA content. For the quantification of DNA IOD, values were evaluated by comparison with those from cells with the known DNA content. Therefore, the DNA content is expressed in a “c” scale in which 1c is half (haploid) of the nuclear DNA content in cells from a normal (non-pathological) diploid population in the G0/G1 cell cycle phase. Non-stimulated porcine lymphocytes were used as standards.

Mitosis in Feulgen stained slides was viewed at 400× magnification. More than 30 experiments and at least 20,000 cells were examined.

### 2.10. DNA Quantification

In order to measure DNA content (in conventional units) by image scanning cytometry, computer-equipped microscope-cytometer SMP 05 (OPTON) was used at 575 nm wavelengths and at 1250× magnification. DNA content was expressed on a “c” scale, in which 1 c is the haploid amount of nuclear DNA that occurred in normal (non-pathologic) diploid populations in the G0/G1 phase. The DNA content of unstimulated swine lymphocytes was used as a diploid standard for measurements. DNA measurements identify nuclei as aneuploid if they deviate more than 10% from 2 c, 4 c, 8 c, or 16 c; i.e., if they are outside of 2 c ± 0.2, 4 c ± 0.4, 8 c ± 0.8, or 16 c ± 1.6 values. 

The variability of DNA content in unstimulated lymphocytes did not exceed 10%. 

### 2.11. ELISA Analysis

Protein levels of cyclin A and cyclin E were measured in PAM culture lysates using sandwich ELISA kits (MyBioSource, San Diego, CA, USA, MBS747375–Cyclin A; MBS753680–Cyclin E). Virus quantification was performed by detection of ASFV antigen (p72) using an ELISA (INgezim PPA DAS 2.0) kit. All experiments were done with the BioTek Epoch 2 microplate spectrophotometer. 

### 2.12. Gene Expression Analysis by Quantitative Real-Time PCR

To determine ASFV expression in PAM cell lines, total viral RNA/DNA was isolated using the HiGene™ Viral RNA/DNA Prep Kit (BIOFACT) following the manufacturer’s instructions. RNA/DNA samples were then reverse transcribed with a REVERTA-L kit (AmpliSens Biotechnologies). 

Quantitative real-time PCR was performed as previously described [23,24] on an Eco Illumina Real-Time PCR system device (Illumina Inc., San Diego, CA, USA). Each reaction mixture (20 µL) was composed of 10 µL of BioMaster HS-qPCR SYBR Blue (2x) mix (Biolabmix), 1 µL of each specific primer, 4 µL of template DNA, and 5 µL of ddH_2_O. Reactions were carried out in the following conditions: polymerase activation: 95 °C for 5 min, 40 cycles: 95 °C for 15 s, 52 °C for 30 s, and 72 °C for 30 s. Standard curves were created using serial 10-fold dilutions of viral DNA. The fluorescence threshold value (Ct) was calculated using the ECO-Illumina system software. Primers used for amplification were designed based on ASFV Georgia 2007/1 sequence (Gene bank: FR682468.2) genes in FASTA format and ordered from Integrated DNA Technology-IDT (https://www.idtdna.com/pagesas follows (accessed on 11 May 2019): 

Primer details: K196R (Thymidine kinase gene) F: 5′-GCAGTTGTCGTAGATGAAG-3′ and R: 5′-CGAAGGAAGCATTGAGTC-3′(length-18–19 bp, amplicon size-69)

R298L (Serine/threonine protein kinase gene) F: 5′-TCTGAAATGTTCTCGGGAAT-3′, R:.5′-GTGTGGACGATAGGTATGG-3′ (length-19–20 bp, amplicon size-71) 

A240L (Thymidylate kinase gene). F: 5′-TGCGTGGAATACTCATTG-3′ R: 5′-TCGTGTCTGGATTAGGAA-3′, R: (length-18 bp, amplicon size-97 bp), E165R (dUTP nucleotidohydrolase) F: 5′-CCTGACCATATCAACATCCTAA-3′ R: 5′-AATCTACCCTCGCCTCTT-3, (length-18–22 bp, amplicon size-37 bp), F334L (Ribonucleotide reductase) F: 5-CAATCATCAATGTCCTTAC-3′, R: 5′-GAATGTTGGAACTGGTAT-3′ (length-18–19 bp, amplicon size-46) F778R (Ribonucleotide reductase) 5′-TATGAACCTGAACTAAGC-3′, R- AATGACAGTAATAGGAACC-3′ (length-18–19 bp, amplicon size-48 bp).

For alignment of the cDNA plots, Cq values were rescaled after comparing with viral genome copies amounts and modified in absolute amounts along the y-axis for better visualization.

### 2.13. Statistical Analysis

All in vitro experiments were conducted in triplicate. The significance of virus-induced changes was evaluated by a two-tailed Student’s *t*-test for parametric values and a Mann-Whitney u-test for non-parametric values; *p* values < 0.05 were considered significant. SPSS version 17.0 software package (SPSS Inc., Chicago, IL, USA) was used for statistical analyses.

## 3. Results

### 3.1. Cell Cycle Changes in Tissues of ASFV Infected Pig

To investigate the effect of ASF infection on the progression of the cell cycle in vivo conditions, liver samples from infected animals were obtained on the 2nd, 3rd, and 4th days after intramuscular injection. It is well known that hepatocytes keep the ability to reenter cell cycle. It results in another important characteristic of hepatocytes-physiological polyploidy. That is why liver histopathology serves as one of the best indices to detect deviations in the cell cycle [25]. Figure 1A illustrates a visualization of S cells in the healthy livers of three-month-old piglets, and Figure 1B shows S sells in the livers of ASFV infected piglets. The distribution of nuclei of hepatocytes by the ploidy classes provides a significant difference between healthy (Figure 1C) and ASFV infected liver cells (Figure 1D). As follows from Figure 1A (visualized by safranin and indigo-picro-carmine trichromic stain) a minority of hepatocytes were in the S stage (shown by triangle), but on the third day post infection by ASFV, a majority of cells were in the S stage (Figure 1B, S cells arrowed). Nuclei distribution in hepatocytes by the ploidy classes in non-infected (Figure 1C) and infected (Figure 1D) tissues revealed a shift of the histogram of the liver nuclei distributed by the ploidy classes to the right, which indicates the synthesis of DNA in nuclei of hepatocytes under the influence of ASFV. In healthy hepatocytes, 10% of the cells were in S stage (Figure 1C), but after ASFV infection more than 71% of cells were in S stage (Figure 1D). At the same time, polyploid cells began to appear in significant quantities (Figure 1D). 

Similar processes occur in quiescent cells in non-proliferating tissues, such as cardiomyocytes (Figure 1F healthy heart section; Figure 1G porcine heart of third-day post infection).

It is well known that the amounts of ASF virus in internal organs are usually observed one to two days after infection and peak three to four days after infection. Figure 1G presents data describing the viral load in examined organs compared to the spleen on the third day post-infection.

### 3.2. Nuclear DNA Synthesis in Infected PAM

The measurements of DNA content in PAM were performed in order to study the cell cycle changes during ASFV infection in vitro. Changes in the DNA amount in PAM nuclei were evaluated by cell cytophotometry in Feulgen stained cells. This technique provides accurate changes in DNA amounts (starting from 5% of the total nuclear DNA). As followed from Figure 2A in a control population of PAM, only cells with normal DNA distribution are present in one peak, and the DNA histogram corresponded to the normal diploid cell population. During the first 8 h of infection, no difference was found in the amount of nuclear DNA compared to the control (Figure 2B-3 hpi, Figure 2C-8 hpi). The first evidence about the additional synthesis of nuclear DNA occurs after 10 hpi (Figure 2D); however, these data are insignificant (*p* < 0.1). The first significant increase of nuclear DNA in infected cells occurs at 12 hpi (Figure 2E) and continued at 14 hpi (Figure 2F) and 16 hpi (Figure 2G).

It is well known that dexamethasone inhibits DNA synthesis by the induced arrest of the cell cycle in G0/G1 [26,27]. To evaluate the role of cell cycle progress in virus replication, we investigated viral levels under the influence of dexamethasone. Dexamethasone in infected PAMs partially stops the synthesis of DNA under the influence of ASFV (Figure 3A). Dexamethasone-dependent PAM arrest in G0 or G1 phases leads to a significant decrease in the production of ASFV (Figure 3B). The change in viral amount under the influence of dexamethasone was measured by the quantification of the levels of the viral protein-p72 (Figure 3C). These data revealed a significant decrease in protein production, thereby confirming a decrease in viral replication when the cell cycle is blocked in the G0/G1 phases.

### 3.3. ASFV (Arm07) Increases Expression of Cyclin A and Cyclin E in PAM Cells

In our previous works it was shown that the PAM entered the S phase and started DNA synthesis upon exposure to the ASF virus [28]. To study cellular proteins, we measured the levels of cyclin A and E in PAM lysates. In intact PAMs, cyclins are not synthesized. During the first 2 h after infection, the amounts of both cyclins were comparable to the control values. An increase in the synthesis of cyclins was observed at a later time. Infection with the ASF virus leads to the synthesis of cyclin A in PAM lysates (Figure 4A) in significant amounts within three-time periods of 4 hpi (*p* < 0.05). The infection of PAM with ASFV also results in a significant but short-term increase in the content of cyclin E (Figure 4B) at 3 hpi (*p* < 0.05). There was then a rapid decrease in the level of both cyclins to background values. 

### 3.4. Measurement of the Transcriptional Activity of Viral Genes Involved in DNA Synthesis during the Pre-Synthesis of Cellular DNA

According to the accepted classification of ASFV genes [29], DNA replication divides the infection cycle into an early phase, before cellular DNA replication begins, and a late phase, after DNA replication begins. Undoubtedly, only those genes can participate in the regulation of the cell cycle that are expressed before cellular DNA replication. Based on the time required for macrophages to make the transition from the G0 to G1 phase, then from G1 to S phase, we investigated the transcriptional activity of several viral genes probably involved in the regulation of DNA synthesis. Consequently, all targeted genes were guided by two requirements. Their expression must begin before 8 h and their function relates (directly or indirectly) to DNA synthesis. Several genes of ASFV have been implicated in DNA synthesis. One of the most important genes is K196R involved in de novo nucleotide synthesis [8]. Although low levels of K196R transcripts were detected in proliferating VERO cells, we investigated the transcription pattern of this gene in G0 PAM. For better understanding and comparing viral infection particles to genome copies number, were used standard 10 fold dilution of infection virus represented in HADU 50/mL. These dilutions (10, 100 and1000 HADU50/mL) were measured by rtPCR to obtain viral genome copies in an amount that corresponded to the HADU dilutions (Figure 5).

As shown in Figure 6A, an increase in transcripts of K196R was observed at 2, 5, and 8 hpi. The transcription level of R298L was then measured (Figure 6B). This gene is serine/threonine protein kinase. Such enzymes play an important role in the regulation of cell proliferation [9]. We have shown that starting from 6 hpi the transcription of R298L is activated, and it continues until the end of the experiment (9 hpi). Next, the transcription level of A240L was measured (Figure 6C). It is a thymidylate kinase involved in the thymidine 5’-diphosphate synthesis [10]. The measurement of the levels of mRNA (cDNA) of the gene showed transcriptional activity between 5 and 8 hpi in infected PAM. Also, mRNA levels of ASFV ribonucleoside-diphosphate reductase’s large (F778R-Figure 6D) and small (F334L-Figure 6E) subunits [30] were examined in the infected PAM in dynamics of viral infection. The transcriptional activity of F778R was detected between 5 and 8 hpi and for F334L in 8–9 hpi. Therefore, the transcription of the large subunit precedes the synthesis of the small subunit. As shown in Figure 6F, dUTP nucleotidohydrolase (dUTPase) mRNA (E165R) levels in infected PAM were significantly increased in 5–9 hpi. 

The cell cycle is under the control of viral genes (at least partially), since cell cyclins are quickly inactivated, and the viral serine/threonine protein kinase gene begins to actively transcribe starting from 6 hpi.

### 3.5. Changes in Transcriptional Activity of the Viral Genes Involved in DNA Synthesis Depend on the Presence of Nucleotides in Medium

It is well known that nucleotides are essential for many biological activities and are constantly generated de novo in all cells. Increased nucleotide synthesis is required for DNA replication and RNA production to enable protein synthesis as cells proliferate at different stages of the cell cycle, during which these actions are regulated at several levels [31]. Next we studied the transcriptional activity of the same viral genes in a medium containing nucleotides (thus, we simulated metabolic activity of a cell in G1 phase compared to a cell in G0 phase). As shown in Figure 6., the presence of nucleotides in the culture medium significantly changed the transcriptional profile of the viral genes associated with nucleotide metabolism (Figure 7A-K196R, Figure 7B-R298L, Figure 7C-A240L, Figure 7D-F334L, Figure 7E-F778R, Figure 7F-E165R). The activity of all investigated genes is significantly lower in the presence of nucleotides in the culture medium, compared with the control results of viral infection in SAM. Moreover, based on the data obtained, all the studied genes at the early stage of infection (preceding the synthesis of cellular DNA) are transcriptionally inactive.

## 4. Discussion

We have shown that the ASF virus manipulates the cell cycle both in G1 (cells of various pig tissues, for example, liver) and in G0 cells: PAM or porcine cardiomyocytes. At the early stage of infection, the cells exit the G0 phase and enter the G1 phase, followed by a transition to the S phase, and then to the G2. At a later stage, the cells are blocked in the G2 phase and the transition to the M phase is prevented. This manipulation leads to the progression of the cell cycle from G0 to G1 and then the S stage. It helps the virus to acquire the necessary nucleotides from the host cell.

This has been shown both in vitro and in vivo. The increase in DNA content in various cells and tissues in the acute form of ASF, which was previously discussed by us [14,28]. In animals, it is undoubtedly associated with viremia, which is observed by the end of day 1 after infection with virulent strains of the virus and, consequently, results in the infection of internal organs. Virulent strains of ASFV were detected in the inner organs, usually at the same time [32]. Despite the fact that the virus primarily infects various macrophages, other cells are also vulnerable to direct infection with this virus in in vivo experiments [33]. Therefore, we are inclined to consider that one of the main reasons for the accumulation of DNA in the nuclei of various pig cells in the acute form of ASF is direct damage of cells by the virus.

Similar processes take place under in vitro conditions. The PAM cell culture is characterized by initially resting cells in the G0 phase. However, soon after the onset of infection, DNA synthesis begins in the nuclei of these cells. Therefore, in the early stages of infection, the virus unblocks cells from the G0 phase and turns on the synthesis of cellular DNA (starting at 10 hpi). The unblocking of PAMs occurs by activating cellular mechanisms (activation of cyclin synthesis), which is probably one of the earliest cytopathological characteristics of ASF infection. The obtained data showed that cyclin A and E accumulate in an infected cell by 3–4 hpi and then their synthesis is abruptly interrupted. Only viral genes expressed before or immediately at that time can affect the synthesis of these cellular proteins. ASFV’s early gene expression in infected cells is detectable as early as 1 hpi; however, in general, transcripts are abundant at 2 hpi with a plateau in accumulation at 2–6 hpi [29]. However, only two genes have been identified as immediate early genes and can accumulate up to 3 hpi. Those genes are L270L, a member of the multigene family 110, and I215L, a protein similar to the ubiquitin conjugating enzymes [29]. Karger et al. (2019) [34] showed the diffuse distribution of pI215L throughout the cytoplasm and nucleus and suggested that this might be a reflection of the ubiquitination of viral and host proteins. Data described previously by [35] about fermentative characteristics of I215L showed that it acts like a ubiquitin-conjugating enzyme. The authors [35] observed that the ASFV-I215L gene is actively transcribed from 2 hpi via qPCR. Therefore, the expression of this gene coincides with the disappearance of cyclins, and we can assume that it is (I215L) probably involved in the suppression of cyclins synthesis in PAM. It is worthy of note that the duration of G1 varies considerably between different cell types under in vitro conditions. Non-differentiated cells remain in the G1 phase for only two to three h, whereas differentiated cell lines remain 8–12 h or more in G1 [36,37]. This time is consistent with our data (described in results) on the onset of DNA biosynthesis in PAM cells starting from 10 hpi.

We can therefore conclude that the synthesis of cellular DNA started at 10–12 hpi. We cannot exclude the viral component in the synthesis of this DNA; however, undoubtedly due to the huge volume of nuclear DNA synthesis (at least three to four orders of magnitude greater than the synthesis of viral DNA), the revealed increase in the amount of DNA in the cell nucleus has a cellular but not viral origin. DNA synthesis in the nuclei of infected PAM does not lead to mitosis. Despite numerous studies, we were unable to identify single reliable mitosis in PAM under the influence of the ASF virus. As mentioned above, the reliable accumulation of DNA in the nuclei of macrophages at 14–16 hpi (Figure 2) was described. Therefore, we can assume that the activation of the cell cycle occurs only until the G2 phase, and then the cell cycle arrest and the prevention of the transition of cells to the M phase follows. The previous study described by [38] suggests that ASFV protein p17 can cause cell cycle arrest and affect the expression of cyclins, including Cyclin A and Cyclin E. However, the inhibition of the expression of cyclins in PAM occurred earlier than the synthesis of p17. Therefore, the blockage of cells in the G2 phase is at least a two-stage process and begins at an early stage of viral infection. It is known that one of the main functions of ubiquitin is the proteolytic degradation of proteins labeled with polyubiquitin chains (in which subsequent ubiquitin units are attached to the side amino groups of the previous ubiquitin molecule) using the 26S proteasome.

It is known that the ASF virus inhibits the translational activity of an infected cell in the early stages of infection as early as 8 h [39,40]. This explains the lack of an increase in cellular cyclin levels after 6 h from the onset of infection. Nevertheless, ASFV infected cells successfully enter from the G0 to G1 stage of the cell cycle. This happens under the influence of viral proteins.

In the nucleus of an infected cell, DNA synthesis starts at 10–12 hpi, and therefore we investigated the transcriptional activity of virus genes directly involved in DNA synthesis or genes involved in early transcription before replication.

Due to the limited pools of intracellular dNTPs, many large DNA viruses encode enzymes involved in nucleotide metabolism in order to increase the precursor pools of dNTPs required for viral DNA replication [5]. The primary target for ASFV replication is non-dividing macrophages with low levels of dNTPs. This is evidenced by the virus-encoded thymidine kinase (K196R) which is non-essential for virus replication in dividing tissue culture cells, but its deletion dramatically reduces virus replication in macrophages [5,6,11]. Moreover, deletion of the thymidine kinase gene induces complete attenuation of the ASFV II genotype, and activity of this gene is required both for efficient replication in porcine macrophages and for virulence in swine. In other words, for replication in vivo, in primary target cells, wild strains of the virus require activation of this gene [8].

We studied the transcriptional activity of the ASFV thymidine kinase (TK) gene (Figure 6C). TK is an enzyme that catalyzes the conversion of thymidine to thymidine monophosphate and then to thymidine triphosphate. The latter is incorporated into DNA since thymidine can only be incorporated into DNA in a phosphorylated form; thymidine kinase plays a key role in the process of DNA synthesis. Now our data explain at what time thymidine kinase of the virus is required for viral replication in G0 porcine macrophages. 

The ASFV TK gene has been shown to be nonessential for the growth of ASFV in cultured hamster and monkey cells [41,42]. The inactivation of the TK gene in poxviruses and herpesviruses showed the gene to be nonessential for growth in cultured cells [6]. 

According to a study described previously by [43], the R298L belongs to the late genes, so its transcription starts after viral DNA replication. However, our data showed that the expression of R298L starts at 6 hpi and continues for a long time. Also, this difference is most likely caused by different types of cells (actively proliferating VERO and G0 PAMs) used in research. Similar genes in other viruses stimulate the infected cell cycle to support viral DNA synthesis [44]. Therefore, we can assume similar functions of the genes A240L (Figure 6A) and R298L (Figure 6B). Ribonucleoside diphosphate reductase (genes F778R Figure 6D and F334L Figure 6E), is an enzyme that catalyzes the formation of deoxyribonucleotides and its blocking inhibits DNA synthesis [5]. The E165R (Figure 6F) gene encodes an enzyme that is involved in pyrimidine metabolism. 

The presence of nucleotides in the culture medium dramatically changed the transcription profile of viral genes associated with nucleotide metabolism, completely turning them off at an early stage of viral infection. 

Summarizing the data obtained on the change in the transcriptional activity of the virus genes in the presence of nucleotides, we can conclude that the genes responsible for the synthesis of nucleotides and the genes affecting the cell cycle behave differently.

The genes affecting the cell cycle simply remain disabled until the beginning of cellular DNA synthesis (8–9 hpi), and the genes responsible for the synthesis of nucleotides in the presence of nucleotides are turned on later and their transcriptional activity is lower than that during virus replication in an environment without nucleotides.

The obtained data with the highest degree of probability can be interpreted as manipulation of the cell cycle by the virus in order to gain access to the nucleotides synthesized by the cell.

By blocking cells (PAMs) in G0 phase, we deprive the virus of the source of nucleotides necessary for the synthesis of viral DNA. Earlier, it was shown that glucocorticoids, and in particular dexamethasone, are able to block the cell cycle in cells of monocytic origin [45,46]. It is well known that glucocorticoids inhibit the proliferation of various types of cells, regardless of their origin. In vitro studies in a variety of cell lines have indicated that glucocorticoids produced a reversible G1-block in the cell cycle. The analysis of DNA content revealed that glucocorticoids arrest cells before S-phase [26,27]. Indeed, the influence of the dexamethasone on infected PAM partially stops an increase of cellular DNA and this coincides with a decrease in the level of viral replication.

Thus, our data revealed that the ASF virus manipulates the cell cycle in infected cells, first by unblocking cells from the G0 phase and sequentially transferring them to the G1 and S phases, and then blocking them in the G2 phase. At least one of the reasons for such manipulation is the need for the virus to synthesize nucleotides by an infected cell. Our assumption is also supported by the data that the activation of the viral thymidine kinase gene occurs in G0 cells and is absent in actively proliferating cells. Overall, we can conclude that the S phase of the cell cycle seems to provide a comfortable environment for successful ASFV replication and completion of the infection cycle (Moore et, al). This phenomenon is also observed in some DNA viruses e.g., herpesviridae [47].

## Figures and Tables

**Figure 1 viruses-14-01593-f001:**
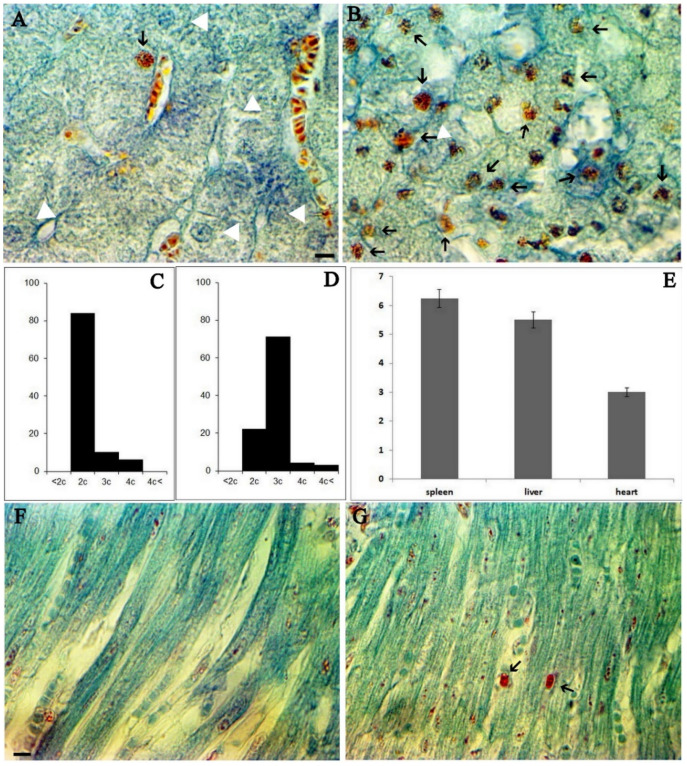
Cell cycle changes in porcine tissue during ASFV infection. (**A**). Healthy porcine liver stained by safranin and indigo-picro-carmine; hepatocyte nuclei in G1 phase shown by white triangles, a single nucleus in S stage shown by a black arrow. (**B**). ASFV infected (third day post infection) porcine liver stained by safranin and indigo-picro-carmine. Many cell nuclei are in the S phase (arrowed). Scale bar 10 µm. (**C**). Hepatocytes nuclei distribution by the ploidy classes in the non-infected liver. (**D**). Hepatocytes nuclei distribution by the ploidy classes in the infected porcine liver (third day post infection). (**E**). The amount of ASFV (suspension of 1 g of tissue) in the spleen, liver, and heart on the third day post infection (mean ± SD). (**F**). Healthy porcine cardiomyocytes stained by safranin and indigo-picro-carmine; absence of cells in S stage. (**G**). ASFV infected (third day post-infection) porcine cardiomyocytes stained by safranin and indigo-picro-carmine; Two cell nuclei are visible in S phase (arrowed). Scale bar 10 µm.

**Figure 2 viruses-14-01593-f002:**
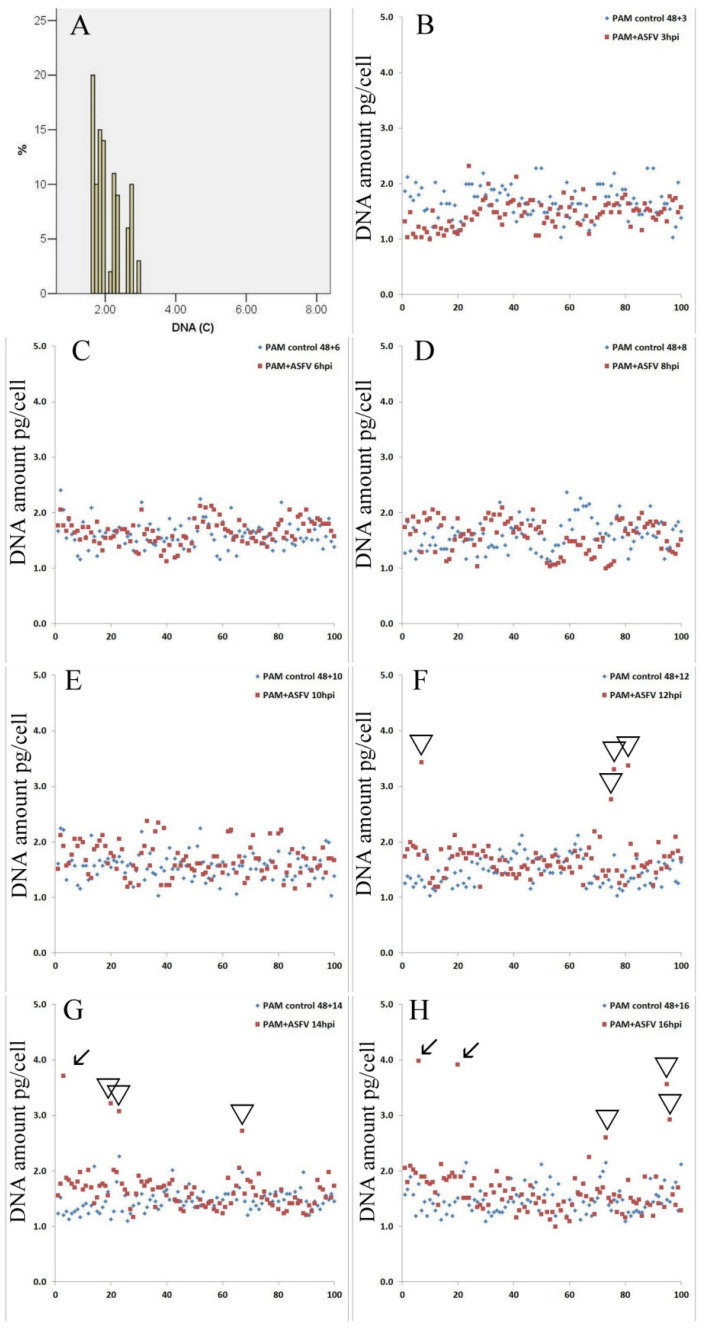
Dynamics of nuclear DNA synthesis in ASFV infected PAM. (**A**). DNA histograms of the intact PAMs. (**B**). DNA amount per cell in pictogram in control and under the ASFV infection (3 hpi). (**C**). DNA amount per cell in pictogram in control and following the ASFV infection (6 hpi). (**D**). DNA amount per cell in pictogram in control and following the ASFV infection (8 hpi). (**E**). DNA amount per cell in pictogram in control and following the ASFV infection (10 hpi). (**F**). DNA amount per cell in pictogram in control and following the ASFV infection (12 hpi). (**G**). DNA amount per cell in pictogram in control and following the ASFV infection (14 hpi). (**H**). DNA amount per cell in pictogram in control and following the ASFV infection (16 hpi). Cells with DNA proliferation in S (marked with triangle) and G2 phase of the cell cycle (marked with arrow).

**Figure 3 viruses-14-01593-f003:**
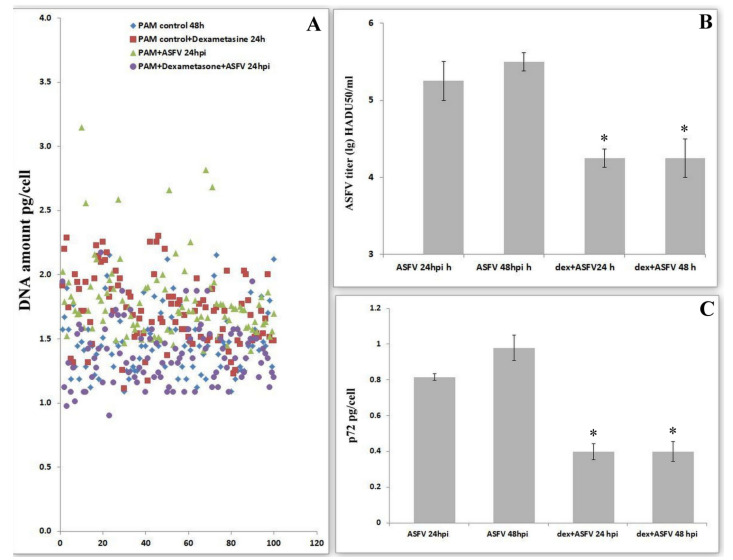
Effects of dexamethasone on the amount of DNA and the replication activity of the ASF virus. (**A**). DNA amount per cell in pictogram in control cells, after dexamethasone treatment and the ASFV infection 24 hpi. (**B**). Effects of dexamethasone on ASFV titers after 24 hpi in PAM; * significant compared to standard infection (*p* < 0.05) (mean ± SD). (**C**). Effects of dexamethasone on ASFV p72 after 24 hpi in PAM; * significant compared to standard infection (*p* < 0.05) (mean ± SD).

**Figure 4 viruses-14-01593-f004:**
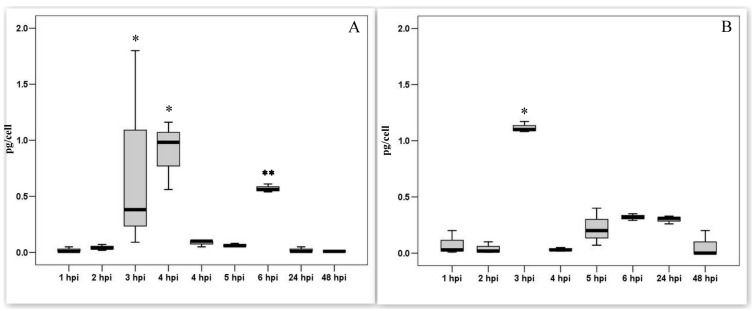
Cyclin A and cyclin E levels in PAM lysates in dynamics of ASFV infection. (**A**). Cyclin A levels in PAM lysates; (mean ± SD); * significant compared to all other measurements (*p* < 0.01–0.05); ** tendency (*p* < 0.1). (**B**). Cyclin E levels in PAM lysates; * significant compared to all other measurements (*p* < 0.01) (mean ± SD).

**Figure 5 viruses-14-01593-f005:**
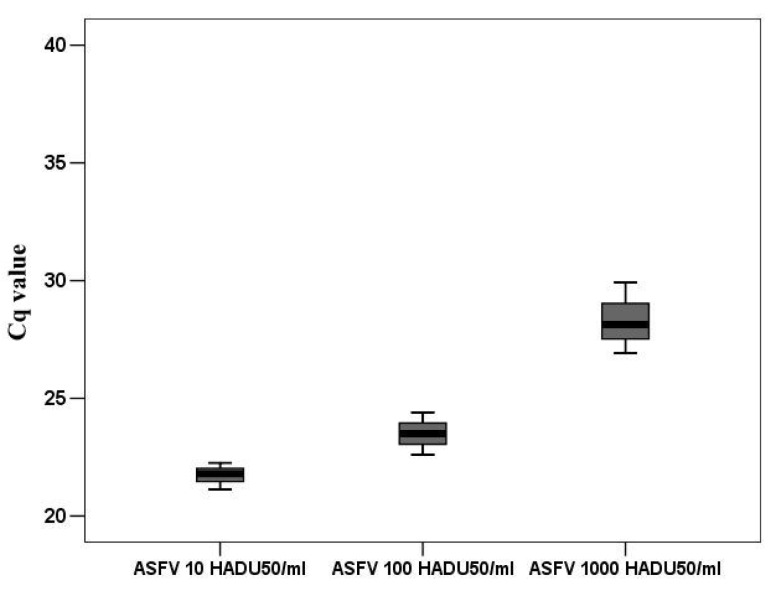
The Cq values of ASFV DNA corresponded to 100, 1000, and 10,000 HADU_50/mL_.

**Figure 6 viruses-14-01593-f006:**
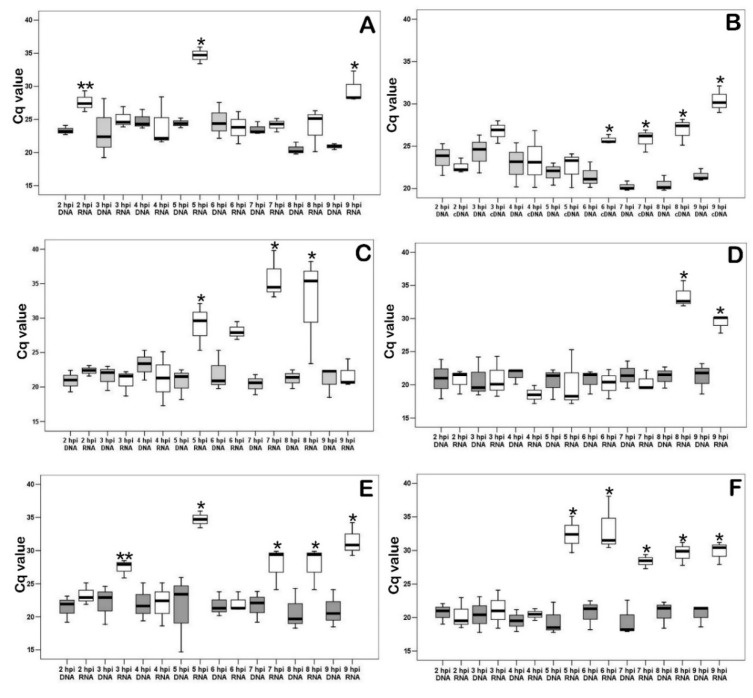
Quantitative real-time PCR results of ASFV K196R, R298L, A2410L, F778R, F334L and E165R mRNA (cDNA) in ASFV infected PAM lysates. (**A**). ASFV thymidine kinase mRNA (K196R) levels in infected PAM in dynamics of viral infection; * significant compared to all another measurements (*p* < 0.01). (**B**). ASFV serine/threonine protein kinase mRNA (R298L) levels in infected PAM in dynamics of viral infection * significant compared DNA levels (*p* < 0.05–*p* < 0.01). (**C**). ASFV thymidylate kinase mRNA (A240L) levels in infected PAM in dynamics of viral infection * significant compared DNA levels (*p* < 0.05–*p* < 0.01). (**D**). ASFV ribonucleoside-diphosphate reductase mRNA large subunit (F778R) levels in infected PAM in dynamics of viral infection * significant compared DNA levels (*p* < 0.05–*p* < 0.01). (**E**). ASFV ribonucleoside-diphosphate reductase mRNA small subunit (F334L) levels in infected PAM in dynamics of viral infection * significant compared DNA levels (*p* < 0.05–*p* < 0.01); ** tendency (*p* < 0.1). (**F**). ASFV dUTP nucleotidohydrolase (dUTPase) mRNA (E165R) levels in infected PAM in dynamics of viral infection * significant compared DNA levels. White boxplots are cDNA and gray boxplots are DNA. DNA and cDNA amounts are graphed on the x-axis. Cq values are graphed on the y-axis.

**Figure 7 viruses-14-01593-f007:**
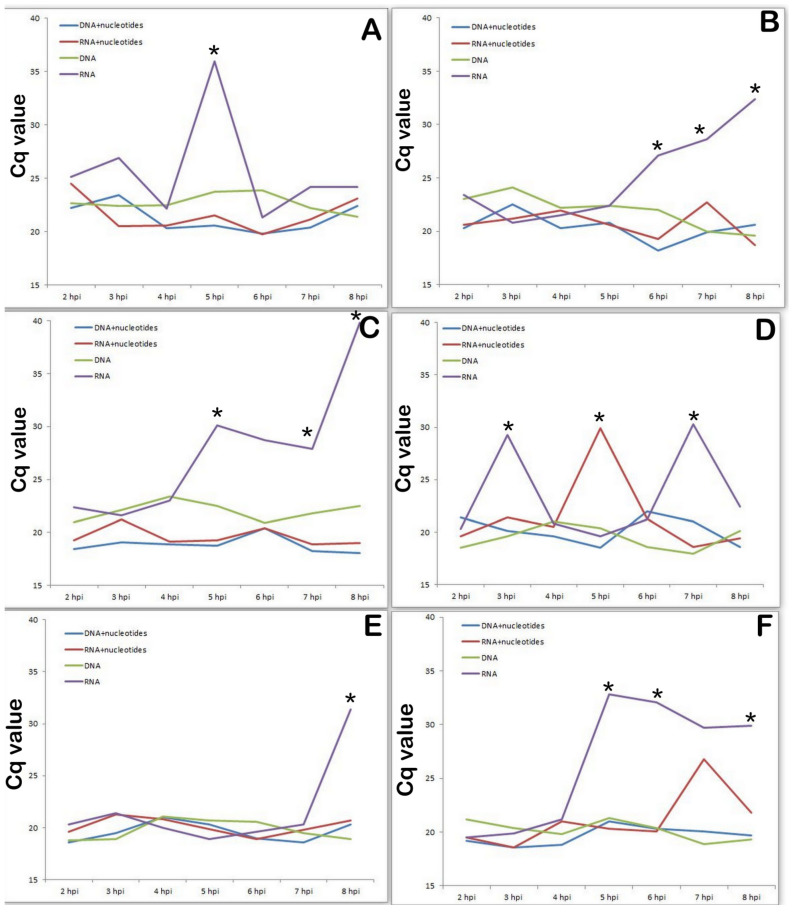
Dynamics of transcriptional activity of ASFV K196R, R298L, A2410L, F334L, F778R and E165R genes mRNA (cDNA) levels in ASFV infected PAM lysates compared with those in nucleotide contained medium. (**A**). Dynamics of transcriptional activity of ASFV K196R in presence of nucleotides in cultural medium. (**B**). Dynamics of transcriptional activity of ASFV R298L in presence of nucleotides in cultural medium. (**C**). Dynamics of transcriptional activity of ASFV A240L in presence of nucleotides in cultural medium. (**D**). Dynamics of transcriptional activity of ASFV F334L in presence of nucleotides in cultural medium. (**E**). Dynamics of transcriptional activity of ASFV F778R in presence of nucleotides in cultural medium. (**F**). Dynamics of transcriptional activity of ASFV E165R in presence of nucleotides in cultural medium. * significant compared with transcriptional activity levels in samples with nucleotides in medium (*p* < 0.05–*p* < 0.01).

## Data Availability

Data are available with this article.

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
