# Peer review of "African Swine Fever Virus Manipulates the Cell Cycle of G0-Infected Cells to Access Cellular Nucleotides"

_viruses, 2022, doi:10.3390/v14081593_

Round 1

Reviewer 1 Report

Thank you for revising the manuscript. Many of the comments were implemented, but not all. For example, the discussion section still lacks direct reference to the results in many places. In addition, not all changes were marked in the manuscript, making it difficult to review again.

Author Response

Q: For example, the discussion section still lacks direct reference to the results in many places.

A: Direct reference was added to the results in the Discussion section.

Reviewer 2 Report

The authors have improved the manuscript but nevertheless, there are still some typos and grammar that need to be corrected to enhance understanding of some sentences.

There is still a lack of clarity with the methodology in terms of the study design. If it's possible provide a sketch so that one can see and understand.

The subtopics in the results section should align with ones under materials and methods so that we see the design and see the result on the other hand. 

Author Response

Point: There is still a lack of clarity with the methodology in terms of the study design. If it's possible provide a sketch so that one can see and understand.

Respond: Corresponding corrections have been made in Materials and methods.

Reviewer 3 Report

1. Please clarify your hypothesis at end of introduction.

2. Add one sentence in the first paragraph of Discussion about the consequence of ASFV cell cycle manipulation.

Author Response

Point 1: Please clarify your hypothesis at end of introduction.

Respond 1: Corresponding data was added.

Point 2: Add one sentence in the first paragraph of Discussion about the consequence of ASFV cell cycle manipulation.

Respond 2: Sentence was added in the mentioned section.

Reviewer 4 Report

Although the work in this paper is interesting, the paper is badly written, lacks information and the authors have not explained their conclusions sufficiently.

The histology work is very convincing showing the cell cycle changes. In figure 1, the histology pictures should be bigger to be able to see the details more clearly and graph G should be smaller. E and F are lacking arrows. Also, all the work on the image cytometry and DNA quantification is good.

Figure 4 shows the ELISA results, which again is convincing – take out ‘White boxplots are cDNA, and gray boxplots are DNA’ from legend as this is measuring protein not DNA.

The paper seems to have been submitted still in ‘review’ and still shows the corrections (in red) and has not been proofread sufficiently and so the text is different sizes, fonts and line spaces. This needs to be corrected.

Questions:

1 – For your qPCRs do you DNase treat? How do you guarantee you only transcribe messenger RNA to cDNA (otherwise you cannot say that this protein is expressed)? Did you carry out all the extractions/qPCRs on mock infected cells? Why did you carry out qPCRs on the extracted DNA? Why carry out a standard curve but not use it quantify your samples?

In the legend for figure 5 you write ‘The Cq values of ASFV DNA corresponded 330 to 100, 1000, and 10000 HADU50/ml.’ Please explain what you mean by this.

In your methods you write ‘For alignment of the cDNA plots, Cq values were rescaled after comparing with viral genome copies amounts and modified in absolute amounts along the y-axis for better visualization. Please explain.

‘Authors observed that 406 ASFV-I215L gene is actively transcribed from 2 hpi via qPCR’ – where is this evidence?

2 – how many replicants did you carry out for your cyclin ELISA? Why do you think that 3 hpi for cyclin A you had such a large variation in values?

3 – In your abstract you say that ‘DNA synthesis in infected alveolar macrophages starts at 10-12 hours post infection’, however, other groups have shown that DNA replication occurs 6 hours post infection (Dixon, et al #6 in your refences). Your own qPCR results for K196R, R298L, A2410L, F778R, F334L and 329 E165R mRNA (cDNA) in ASFV infected PAM lysates show an increase of RNA at 5 or 6 hpi. Please clarify this point.

There are too many mistakes and comments to list all of them, but the following should be addressed:

1 – All graphs should include a Y and X axis title so the reader can follow without having to go back to the methods. Make the axis legible (for example, figure 5 the X axis on the graphs is too small)

2 – Be consistent with terms and values (for example, HAD50/ml or HADU50/ml)

3 – Be careful with repeating what you have previously said. For example, in the methods section 2.10 is practically a repeat of 2.09

4 – Be careful with the written text (e.g. don’t start a sentence with ‘because’, miss out words such as ‘and’ and ‘the’, miss out spaces between words, etc)

5 – Sometimes, the meaning of sentences/paragraphs are not very well written and could be written better to make the point clearer.

Author Response

Point 1: The histology work is very convincing showing the cell cycle changes. In figure 1, the histology pictures should be bigger to be able to see the details more clearly and graph G should be smaller. E and F are lacking arrows. Also, all the work on the image cytometry and DNA quantification is good.

 Respond 1: Corresponding corrections were done in the paper.

Point 2: Figure 4 shows the ELISA results, which again is convincing – take out ‘White boxplots are cDNA, and gray boxplots are DNA’ from legend as this is measuring protein not DNA.

Respond 2: These data have been added in the paper.

Point 3: For your qPCRs do you DNase treat? How do you guarantee you only transcribe messenger RNA to cDNA (otherwise you cannot say that this protein is expressed)? Did you carry out all the extractions/qPCRs on mock infected cells? Why did you carry out qPCRs on the extracted DNA? Why carry out a standard curve but not use it quantify your samples?

Respond 3: No, we didn't use DNAase as we wanted to measure transcript levels in cells by comparing DNA and cDNA levels before and after reverse transcription. Is it the result of increased amount of amplified viral genes with low transcription or highly increased transcription in limited copies of ASFV genes? Standard curve is also applied for sample quantification. We also carried out study on mock infected cells. 

Point 4: In the legend for figure 5 you write ‘The Cq values of ASFV DNA correspondinged 330 to 100, 1000, and 10000 HADU50/ml.’ Please explain what you mean by this.

Respond 4: Corresponding corrections were done in the paper.

Point 5: In your methods you write ‘For alignment of the cDNA plots, Cq values were rescaled after comparing with viral genome copies amounts and modified in absolute amounts along the y-axis for better visualization. Please explain.

Respond 5: According to the standard method higher Cq values are determined as low cDNA concentration, and vice versa, so transcription values of Cq were increased (rescaled version) comparing with HADU plots.

Point 6: ‘Authors observed that 406 ASFV-I215L gene is actively transcribed from 2 hpi via qPCR’ – where is this evidence?

Respond 6: This is not our data, so the corresponding reference was added.

 Point 7: How many replicants did you carry out for your cyclin ELISA? Why do you think that 3 hpi for cyclin A you had such a large variation in values?

Respond 7: This is our data and such variation in values may depended on the phenomenon that some cells are not infected by the virus. Corresponding data was added.

Point 8: In your abstract you say that ‘DNA synthesis in infected alveolar macrophages starts at 10-12 hours post infection’, however, other groups have shown that DNA replication occurs 6 hours post infection (Dixon, et al #6 in your refences). Your own qPCR results for K196R, R298L, A2410L, F778R, F334L and 329 E165R mRNA (cDNA) in ASFV infected PAM lysates show an increase of RNA at 5 or 6 hpi. Please clarify this point.

Respond 8: Dixon and coauthors showed data on G1 cells (VERO cell line). Our data performed on G0 cells (PAMs) and transcription of the viral genes was observed before the cell DNA synthesis in all cases.

Point 9: Be consistent with terms and values (for example, HAD50/ml or HADU50/ml)

Respond 9: Corresponding corrections were done in the Main document.

Point10:

1 - All graphs should include a Y and X axis title so the reader can follow without having to go back to the methods. Make the axis legible (for example, figure 5 the X axis on the graphs is too small)

2 – Be consistent with terms and values (for example, HAD50/ml or HADU50/ml)

3 – Be careful with repeating what you have previously said. For example, in the methods section 2.10 is practically a repeat of 2.09

4 – Be careful with the written text (e.g. don’t start a sentence with ‘because’, miss out words such as ‘and’ and ‘the’, miss out spaces between words, etc)

5 – Sometimes, the meaning of sentences/paragraphs is not very well written and could be written better to make the point clearer.

Respond 10: Corresponding corrections were done in the paper.

Round 2

Reviewer 4 Report

Thank you for addressing all my comments and questions.

This manuscript is a resubmission of an earlier submission. The following is a list of the peer review reports and author responses from that submission.

Round 1

Reviewer 1 Report

The Authors described some  of the activities that takes place during infection within the cell and is surely and addition to what we know about this virus. However, the manuscript has some problem with language in terms of clarity, conciseness and spellings that needs to be cross-checked and corrected to enhance understanding.

Some statements like "..30 experiments and at least 20,000 examine cells" has not connection with Materials and Methods.

The authors also mentioned "chapter" and needs to clarify. 

Reviewer 2 Report

The manuscript by Avagyan and Colleagues is a descriptive approach to analyze the abundance of different genes of ASF in porcine alveolar macrophages and in three months old piglets. The authors describe an altered cell cycle after infection and examined six ASFV genes by real-time PCR over time after infection.

General remarks:

It is obscure when and for what purpose which experiment was performed. As a result, the confirmability suffers considerably. In some cases, the choice of methods does not appear optimal for collecting the data presented here. Transcriptional activity should be confirmed by other methods (e.g. RNA Seq). Furthermore, it would be nicer to display the graphics in a uniform way. In some text passages, the manuscript has linguistic deficiencies and the style does not correspond to that of a scientific publication (e.g. “So the genes we studied”, “in addition to everything“).

Abstract:

It would be helpful to include the functions of the genes.

Introduction:

Can you please comment on the gene nomenclature?

Materials & methods?

How big are the target genes? How long are the generated amplicons?

Results:

3.1.: Does that apply for all days after infection or only for the piglets euthanaized at 3dpi?

3.2., second paragraph: Was this shown or is this an assumption?

3.2., third paragraph: Could you please comment on the mode of action of dexamethasone?

3.3., first paragraph: Reference? Infection of AMP? Do you mean PAM?

3.4.: References missing for first paragraph. Are you really measuring transcriptional activity? As fas as I understood or as you describe in the method section, your extracting DNA as well as RNA followed by a RT-qPCR. In order to measure transcriptional activity, you would need to measure directly RNA transcripts by for example transcriptomics or reporter assays. How can you be sure to measure only mRNA? And how can you be sure that dexamethasone blocks PAMs in G0? You describe an increase in R298L transcription activity starting from 6 hpi. An increase would mean a lower Ct value but in Fig 5C higher Ct values are depicted over time. Wouldn’t this mean a decrease of this gene? Same for the other figure parts.

3.5.: In what kind of experimental setting did you study the effect of nucleotides? The different media composition is not mentioned in the materials&methods section.

Fig 1A: White triangles not visible. G: to which sample volume do the specifications refer? mg?

Fig 2: Legend illegible. The differences in DNA content per cell are not clear in this figure. Maybe it helps to change the scaling. Otherwise, this figure could be moved to the supplement.

Fig 4: Is pg/cell correct?

Fig 5: What are the grey and white boxes? An explanation is missing. It would be helpful ton include the name of the measured genes into the plots. A. You mention in the text showing and increase of K196R transcripts over time. This does not match the depicted figure. i.e. C: Is this data for R298L or A240L? According to the legend and the description in the text it should be A240L but the description of an increase matches more the explanation to R298L.

Fig 6: What does RNA or DNA mean? What was measured?

Discussion:

The reference to the results is missing, which makes the discussion incomprehensible.

Reviewer 3 Report

The work by Avagyan et al. described how ASFV manipulate cell cycle. Authors carried out both in vitro work on porcine alveolar macrophages (PAM) and in vivo work, analyzing cell cycle changes in hepatocytes and cardiomyocytes. Researchers described that the virus unblocks PAM from the GO phase to the S phase and later blocks these cells in the G2 phase. DNA synthesis starts 10-12 h pi and before several ASFV genes are activated. The work address an interesting point because the interaction of ASFV with its target cells (macrophages) still present several gaps.

Nevertheless, I think that the work need substantial revision before publication.

Introduction:

  • Several sentences are not referenced.

African swine fever virus (ASFV) is the only species in the genus Asfivirus, family Asfarviridae, and order Asfuvirales. It is a large double-stranded DNA virus’.

‘Moreover, the main research in this direction was carried out on cell cultures, i.e. on actively proliferating cells, while the main target for the ASF virus is differentiated macrophages, which normally belong to G0 cells’.

  • This sentence need to be rephased: ‘viral replication effects may not necessarily be the same during natural infection in normal for their replication G0 cells (porcine macrophages) and in actively proliferated cells.’

Material and methods

  • Not cleat which animals used for in vitro experiments and which for in vivo.
  • How viral stocks were produced and titers determined + references.
  • Not clear which were the in vivo experiments and which were the in vitro.
  • This should stay in a separate subchapter: ‘Infected pigs were euthanized in pairs on the second, third, fourth, sixth, and seventh day post-infection (dpi). During necropsy, the inner organs were carefully removed and fixed in 10% buffered formalin solution (pH 7.2) for histopathology studies’
  • How DNA was quantified, with which instrument?
  • ELISA: instrument used to read OD.

Results

  • This sentence need to be references: ‘It is well known that hepatocytes keep the ability to reenter the cell cycle. It results in another important characteristic of hepatocytes - physiological polyploidy. That is why liver histopathology serves as one of the best indices to detect deviations in the cell cycle.’
  • 2 Feulgen staining is not described in materials and methods.
  • 2 Significant… are the stats described in the figures?
  • 3 ‘Previously, it was shown that upon exposure to ASF virus, PAM entered S phase of the cell cycle and started DNA synthesis.’ … where? In which work??

Discussion

It is unclear which were the results from the in vivo work and which from the in vivo.

Overall, results should be better discussed.

Maybe it is useful to describe potential implication in vivo of your discovery.

Figure Legends

1 à data are mean + SD? How many replicates?

3 à data are mean + SD?

4 à data are mean + SD? * meaning and ** meaning

5 à box and whisker plots what do they represent? * is p < 0.05 or p < 0.0.1? Use different symbols. ** use different symbols for tendency.